# GeomVerse
# A Systematic Evaluation of Large Models for Geometric Reasoning

**Mehran Kazemi** [1]  **Hamidreza Alvari** [1]  **Ankit Anand** [1]  **Jialin Wu** [1]  **Xi Chen** [1]  **Radu Soricut** [1]

## Abstract

Large language models have shown impressive results for multi-hop mathematical reasoning when the input question is only textual. Many mathematical reasoning problems, however, contain both text and image. With the ever-increasing adoption of vision language models (VLMs), understanding their reasoning abilities for such problems is crucial. In this paper, we evaluate the reasoning capabilities of VLMs along various axes through the lens of geometry problems. We procedurally create a synthetic dataset of geometry questions with controllable difficulty levels along multiple axes, thus enabling a systematic evaluation. The empirical results obtained using our benchmark for state-of-the-art VLMs indicate that these models are not as capable in subjects like geometry (and, by generalization, other topics requiring similar reasoning) as suggested by previous benchmarks. This is made especially clear by the construction of our benchmark at various depth levels, since solving higher-depth problems requires long chains of reasoning rather than additional memorized knowledge.

## 1. Introduction

Multi-hop reasoning is a fundamental element in intelligence: it allows us to combine multiple pieces of information to answer questions or solve problems. While formal reasoning such as automated theorem proving (Robinson, 1965; Kovács & Voronkov, 2013; Schulz, 2002) has been a key focus in the AI literature, recent years have witnessed a great amount of progress in multi-hop reasoning with natural language thanks to the advances in pre-trained large language models (LLMs) (Wei et al., 2022; Nye et al., 2022; Kazemi et al., 2023b; Saparov et al., 2023; Yao et al., 2023;

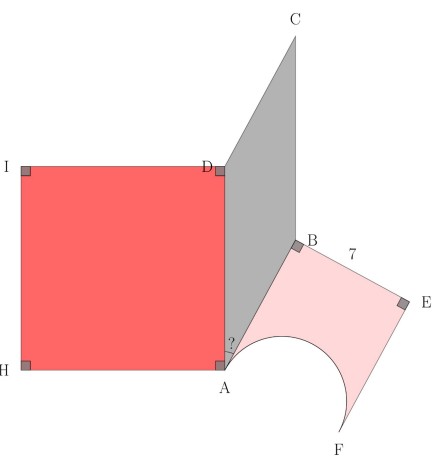

Figure 1: Sample GeomVerse problem. **Question:** *If the ABEF shape is a rectangle where a semi-circle has been removed from one side of it, the perimeter of the ABEF shape is 34 [...] compute the degree of the DAB angle. Assume $\pi = 3.14$. Round computations to 2 decimal places.* **Solution:** *The diameter of the semi-circle in the ABEF shape is equal to the side of the rectangle with length 7 so the shape has two sides with equal but unknown lengths, one side with length 7, and one semi-circle arc with diameter 7. So the perimeter is $2 * UnknownSide + 7 + \frac{7\pi}{2}$ [...] the length of the AB side is $\frac{16.01}{2} = 8$. [...] the final answer is 28.69.*

Pan et al., 2023). Among various types of multi-hop reasoning, mathematical reasoning has turned into a key focus domain for AI researchers (Lu et al., 2022; Lewkowycz et al., 2022) with many recent works targeting to solve open problems in mathematics (Fawzi et al., 2022; Davies et al., 2021). It is an appealing domain for AI research due to various reasons: it is a primitive skill that is essential for many tasks, it has an open-ended nature, and due to various challenges such as limited data it still remains a challenge for LLMs and modern AI systems. Recently, the International Math Olympiad (IMO) grand challenge (Selsam et al., 2020) was announced where the goal is to build an AI system that can win a gold medal in IMO, one of the most prestigious competitions. Not only research, with advancements in LLMs, many new applications and products are

[1]Google DeepMind. Correspondence to: Mehran Kazemi <mehrankazemi@google.com>.

*The first AI for MATH Workshop at the 41st International Conference on Machine Learning*, Vienna, Austria. Copyright 2024 by the author(s).

leveraging AI research for education to build personalized tutors (Abdelghani et al., 2023; Khan, 2023). One of the key challenges so far has been to improve the performance of these systems in STEM subjects.

Due to the vast popularity of mathematical problem solving both from research and product perspectives, several datasets have been developed for measuring and improving the mathematical reasoning of LLMs (Cobbe et al., 2021; Ling et al., 2017; Hendrycks et al., 2021) and are widely adopted by the research community. While existing datasets mostly focus on textual problems, there are several bodies of mathematical problems that require both textual and visual understanding of the problem. Being one of the main school curriculum and having a high presence in many math competitions including IMO, *geometry* is a key domain in this space. With the fast pace in adoption of the vision-language models (VLMs) (Chen et al., 2022b; OpenAI, 2023) in various aforementioned applications, it is crucial to measure and improve their performance on such problems. Previous work has created a number of datasets with geometry questions based on high-school, college, or SAT exams, and developed specific models for this task. While evaluating VLMs on such datasets may provide a holistic understanding of the general capability of the models, such evaluation may provide little information about the specific areas of strengths and weaknesses of VLMs and hence provide little guidance on where research should focus. Recent years have witnessed a surge of interest in synthetic datasets that allow for a systematic evaluation of the boundaries of capabilities and the limitations of the state-of-the-art models (see, e.g., (Lindström & Abraham, 2022; Borisov et al., 2022; Kazemi et al., 2023a; Gekhman et al., 2023; Vaska & Helus, 2023; Fatemi et al., 2024)).

In this paper we create GeomVerse, a dataset of synthetically generated geometry questions that require multi-hop mathematical reasoning over text and image. We bridge reasoning about geometry problems and logical reasoning, allowing us to measure model performances on reasoning factors that may go beyond geometry and may be present in many (mathematical) reasoning problems on text and image. In other words, GeomVerse allows for unveiling the reasoning ability of VLMs across several axes, by using geometry as a lens. We also measure model performances on geometry-specific axes of difficulty. This enables a systematic evaluation of VLMs on this task. A sample generated problem and solution can be viewed in Figure 1.

Some of the main findings from our systematic evaluation on GeomVerse are summarized below. Firstly, through the unique property of GeomVerse that allows for constructing benchmarks at various depths, we find that current VLMs are not as capable in subjects like geometry as suggested by previous benchmarks, showing that they may still be immature for product applications such as AI tutoring. Importantly, since several of the difficulty axes we study are not specific to geometry, our results reveal a number of important failure modes as well as a significant gap in the reasoning capacity of state-of-the-art VLM that may go beyond geometry. Secondly, finetuning VLMs to produce the entire solution substantially improves their performance for in-distribution problems but that does not generalize to out-of-distribution problems. Thirdly, VLMs struggle more with increasing in depth rather than width of reasoning. And fourthly, VLMs are rather robust to the question and image representation.

## 2. Related Work

Our work is related to several research directions in the literature as summarized below.

**Vision-Language Models (VLMs):** Recent VLMs (Chen et al., 2022b; Allaway et al., 2022; Alayrac et al., 2022; Li et al., 2023; Wang et al., 2022; Chen et al., 2023) have demonstrated promising performance on a wide range of image and video tasks including captioning, question answering and visual reasoning. However, the capabilities of performing multi-modal multi-hop (mathematical) reasoning are less investigated. Because these VLMs are generative black-boxes, understanding how well they can comprehend and answer the multi-hop questions is a critical topic.

**Multi-Hop Reasoning Datasets:** There are a number of datasets available in the literature that require multi-hop logical (Tafjord et al., 2021; Kazemi et al., 2023a; Zhong et al., 2021) and mathematical (Cobbe et al., 2021; Ling et al., 2017; Hendrycks et al., 2021) reasoning over text. Previous work has also developed a number of geometric reasoning datasets (Seo et al., 2015; Lu et al., 2021; Chen et al., 2021; 2022a; Zhang et al., 2023) that require reasoning over both text and image. Table 1 provides an overview of the existing datasets and compares them along four axes: 1- requiring textual understanding, 2- requiring visual understanding, 3- involving mathematical reasoning, and 4- automatic control of the difficulty level (thus allowing for a systematic evaluation).

**Multi-Hop Reasoning Approaches:** Some of the approaches for improving the multi-hop reasoning of LLMs and VLMs range from pre-training on relevant data (Hendrycks et al., 2021; Lewkowycz et al., 2022), finetuning with (Nye et al., 2022; Dalvi et al., 2021; Zelikman et al., 2022; Kazemi et al., 2023a) and without (Clark et al., 2021; Betz et al., 2021) explicitly generating the solution, in-context learning with solutions (Wei et al., 2022), decomposing the problem into smaller pieces and solving them separately (Zhou et al., 2023; Khot et al., 2023) and using

| Dataset → Feature ↓ | ProofWriter (Tafjord et al., 2021) | BoardgameQA (Kazemi et al., 2023a) | AR-LSAT (Zhong et al., 2021) | AQUA (Ling et al., 2017) | GSM8k (Cobbe et al., 2021) | CLEVR-Math (Lindström & Abraham, 2022) | ChartQA (Masry et al., 2022) | GeoS (Seo et al., 2015) | GeoQA (Chen et al., 2021) | Geometry3k (Lu et al., 2021) | UniGeo (Chen et al., 2022a) | PGPS9K (Zhang et al., 2023) | GeomVerse |
|---|---|---|---|---|---|---|---|---|---|---|---|---|---|
| Textual Understanding | ✓ | ✓ | ✓ | ✓ | ✓ | ✓ | ✓ | ✓ | ✓ | ✓ | ✓ | ✓ | ✓ |
| Visual Understanding | ✗ | ✗ | ✗ | ✗ | ✗ | ✓ | ✓ | ✓ | ✓ | ✓ | ✓ | ✓ | ✓ |
| Mathematical Reasoning | ✗ | ∼ | ✗ | ✓ | ✓ | ∼ | ∼ | ✓ | ✓ | ✓ | ✓ | ✓ | ✓ |
| Automatic Difficulty Control | ✓ | ✓ | ✗ | ✗ | ✗ | ∼ | ∼ | ✗ | ✗ | ✗ | ✗ | ✗ | ✓ |

Table 1: A comparison of GeomVerse with some of the recent and/or widely-used multi-hop (logical or mathematical) reasoning datasets. We use ∼ when a dataset contains a property to a limited extent.

LLMs/VLMs as tools within classical algorithms (Kazemi et al., 2023b; Creswell et al., 2023). In the realm of reasoning about geometry problems, existing work typically develops specialized models or tools (e.g, (Trinh et al., 2024)) or resorts to distillation strategies (e.g., (Gao et al., 2023)); measuring the reasoning ability of general-purpose VLMs is less studied.

## 3. The GeomVerse Dataset

We start with some preliminaries and terminologies. Then, we explain how GeomVerse is created. The dataset will be publicly available upon the acceptance of the paper.

### 3.1. Multi-Hop Logical Reasoning

A logical theory consists of facts and rules. Consider the following theory as a running example:

$$Facts : \{a, b\}$$
$$Rules : \{a \Rightarrow c, a \wedge b \Rightarrow d, d \Rightarrow e\}$$

The theory contains two facts specifying $a$ and $b$ are true, and three rules specifying $a$ implies $c$, $a$ and $b$ imply $d$ and $d$ implies $e$. Starting from the facts, one can apply *deduction* on the set of facts and the rules to derive new facts and answer queries (e.g., we can query whether $e$ holds). We define the *depth* of a query as the number of hops of reasoning required to prove it, and the *width* of a query as the maximum number of branches in the proof of the query. For a given query, any fact or rule not necessarily in the proof of the query is referred to as a *distractor*. For example, if we query $a$ both the depth and width are 0, if we query $c$ the depth is 1 and the width is also 1, if we query $d$ the depth is 1 and the width is 2, and if we query $e$ both the depth and the width are 2. When we query $e$, the

rule $a \Rightarrow c$ is a distractor. Note that queries with width 1 correspond to a chain of reasoning, whereas higher width queries correspond to a tree of reasoning.

### 3.2. From Logical to Geometric Reasoning

Geometry problems often provide values for certain elements (e.g., sides, angles, areas). Using geometric rules and formulas, we can deduce the values of the remaining elements one by one. The elements whose values are given to us can be thought of as facts in logical theories, the geometry rules and formulas can be considered as the rules in logical theories, and the process of deriving the hidden values can be thought of as the deduction.

As an example, the solution to the problem in Figure 1 can be formulated in logical form as:

$$Facts : \{A_{AHID}, A_{ABCD}, P_{ABEF}, L_{BE}\}$$
$$Rules : \{A_{AHID} \Rightarrow AD, P_{ABEF}, L_{BE} \Rightarrow L_{AB},$$
$$A_{ABCD}, L_{AD}, L_{AB} \Rightarrow D_{DAB}\}$$

where $A_x$, $P_x$, $L_x$ and $D_x$ represent the area of a shape, perimeter of a shape, length of a side, and degree of an angle respectively. We note two key differences with logical reasoning: 1- unlike in deductive logical reasoning, the rules are not given to the model and the model has to use its own geometry knowledge (learned from pre-training or finetuning) to apply the right geometry formulae and derive new values, 2- in the case of geometry, applying rules involves computations.

### 3.3. Creating the GeomVerse

To create GeomVerse, we fix a set of 12 standard and non-standard shapes $\mathcal{S}$ as demonstrated in Figure 2 and gather a

number of rules/formulas $\mathcal{F}_s$ for each shape $s \in \mathcal{S}$ (e.g., the Pythagorean theorem) with a total of 68 formulas across all shapes. We further use supplementary and complementary angles as two special shapes with only a single formula each. For a formula $f$, we let $f_{in}$ represent the input elements and $f_{out}$ represent the element whose value can be computed based on the formula and the inputs (e.g., for the Pythagorean theorem, the two sides can be the input and the hypotenuse can be the output). Then, similar to several existing works on constructing multi-hop, textual logical reasoning datasets or textual stories (Kazemi et al., 2023a; Ye et al., 2022), we adopt a backward generation strategy where we start by generating a question, and then adding rules to increase the depth and width of reasoning to the desired amount.

**Generating Examples with Depth 1:** To generate an example with depth 1, we can simply sample a shape $s \in \mathcal{S}$ and formula $f \in \mathcal{F}_s$. Then, we let $facts = f^{(in)}$, $query = f^{(out)}$, and with the only required rule in the solution being $rules = \{f\}$.

**Increasing the Depth:** Let $f_1$ be the formula we sampled for the depth 1 example and $f_1^{(in)}$ and $f_1^{(out)}$ be the inputs and output of $f_1$. To increase the depth to 2, we select one of the elements $e$ in $f_1^{(in)}$ and do not provide it in the facts. Instead, we sample a new shape $s_2$ and formula $f_2$ such that $f_2^{(out)}$ has the same type as $e$ and we tie the values of $e$ and $f_2^{(out)1}$. For example, if $e$ is one of the sides of a triangle, then $s_2$ can be a square and $f_2$ can be the formula of deriving the side of a square from its area, where the square and the triangle share the same side. Then $facts = (f_1^{(in)} - e) \cup f_2^{(in)}$, $query = f_1^{(out)}$, and the required rules are $rules = \{f_1, f_2\}$ with $f_2$ providing the value for $e$ and then $f_1$ using this value to answer the query. The depth can be further increased in a similar way by appending a new shape and formula to one of the elements in $f_2^{(in)}$.

**Increasing the Width:** Let $s$ and $f$ be the shape and formula we sampled at some depth for the construction of an example and $e_1$ and $e_2$ be two connectable elements (side or angle) in $f^{(in)}$. We can include only $f^{(in)} - \{e_1, e_2\}$ in the facts, and append new shapes and formulas as explained above so that the values for $e_1$ and $e_2$ can be derived.

**Distractors:** Distractors can be added in a post processing step. Consider a Depth 2 (Width 1) example and suppose $e$ is the element that has to be computed in the first hop and be used in the second hop. If we provide the value of $e$ as input, then the model turns into a Depth 1 problem with a distracting shape and corresponding values. In Figure 1, for example, if we provide the value of the $AD$ side as

---

[1]Note: $e$ should have a type that allows it to be connected to another shape (e.g., side or angle).

---

**Algorithm 1** BackwardGenerate
**Input:** Shared element $e$, Shared element type $\tau$ Depth $d$ [t]
**if** d == 0 **then**
    **do**
        s = RandomSelect($\mathcal{S}$)
        f = RandomSelect($\mathcal{F}_s$)
    **while** $f^{(out)}$.type != $\tau$
    Append $s$ to other shapes on $e$.
    Randomly assign values to $f^{(in)}$.
    Provide $f^{(in)}$ values as facts.
**else**
    **do**
        s = RandomSelect($\mathcal{S}$)
        f = RandomSelect($\mathcal{F}_s$)
        $\mathcal{E}$ = ConnectableElements($f^{(in)}$)
    **while** $f^{(out)}$.type != $\tau$ **OR** $|\mathcal{E}| = 0$
    Append s to other shapes on $e$.
    Randomly assign values to $f^{(in)} - \mathcal{E}$.
    Provide $f^{(in)} - \mathcal{E}$ values as facts.
    $e_1, \ldots, e_m$ = SampleElems($\mathcal{E}, p_{branch}$)
    **for** $e \in \{e_1, \ldots, e_m\}$ **do**
        BackwardGenerate(e, e.type, d-1)

---

input, then the square and its corresponding values can be considered as distractors.

**The Generation Algorithm:** Algorithm 1 adopts the high-level idea of the *GenerateTheory* algorithm from Kazemi et al. (2023a) for recursively generating geometry problems (as opposed to logical theory problems) in a backward fashion. Initially, we select one element type $\tau$ to be asked for in the question (e.g., side, angle, area, perimeter, etc.), and a desired depth $d$. Then we call the *BackwardGenerate* function. If $d = 0$, we sample a shape $s \in \mathcal{S}$ and formula $f \in \mathcal{F}_s$ such that the type of the element in $f^{(out)}$ is $\tau$, append the shape $s$ to the previous shapes on the shared element, assign random values to the elements in $f^{(in)}$ and provide them as facts[2]. Otherwise, we sample a shape $s \in \mathcal{S}$ and formula $f \in \mathcal{F}_s$ such that 1) the type of the element in $f^{(out)}$ is $\tau$ and 2) there is at least one connectable (side or angle) element in $f^{(in)}$. Then, we select a subset $\mathcal{E}$ of the elements in $f^{(in)}$ for expanding the number of hops. If $p_{branch} = 0$, we only select one of the elements from $f^{(in)}$

---

[2]During random value assignment, we test multiple factors to ensure the assigned values are sensible (e.g., the sides of a right triangle are smaller than its hypotenuse) and re-assign values until these criteria are met. Sometimes, this becomes impossible due to some values that are derived from other hops; in these cases, we simply discard the example and generate another example from scratch.

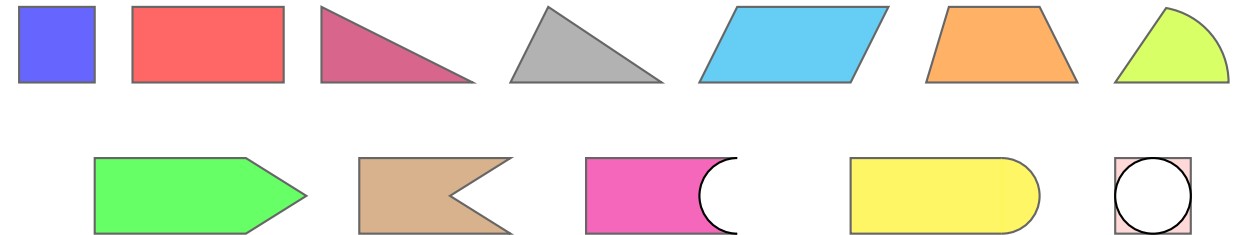

Figure 2: The standard shapes (top row) and non-standard shapes (bottom row) used in our dataset.

which introduces no branching and so no increase in width. Otherwise, with probability $p_{branch}$ we select a second element as well, which leads to a branching and so increases the width. We append the shape $s$ to the previous shapes on the shared element, assign random values to the elements in $f^{(in)} - \mathcal{E}$ and provide them as facts. Then, for each element in $\mathcal{E}$, we recursively call the *BackwardGenerate* function to append new shapes such that these values can be derived. A visual example of the procedure is provided in Appendix C.

**Automatic Question and Solution Generation:** We automatically produce a question that provides the facts as input and asks for $f^{(out)}$ where $f$ is the first formula used. We also keep track of the required rules (including shapes, formulas, and the shared elements) during the generation process (excluded from Algorithm 1 for brevity) and automatically produce a solution by applying deduction and computations on the rules and facts.

**Text-Only vs. Text-Image:** We create two versions of our problems. In one version, all the required information is given in the question and the image is not needed for answering the question (although the presence of the image can make it easier to understand the problem), and in the other version some information is given in the image and some in the question text so both the image and text of the question are required. We use the former to experiment with text-based LLMs and the latter to experiment with VLMs.

**Coverage:** The connection between Algorithm 1 and logical reasoning helps specify what classes of geometry problems are covered by Algorithm 1. Specifically, Algorithm 1 can generate any geometry problem $\mathcal{P}$ containing a tree of shapes where each shape is connected to its parent shape via a single side or a single (vertical) angle, and where the solution can be found by finding the values of the shared elements bottom-up on the tree.

**Further Considerations:** While Algorithm 1 can generate problems with overlapping shapes, to ensure the quality of the generated examples remains high without any human involvement in the generation process, we only accept the generated examples where the shapes are non-overlapping.

### 3.4. Quality Check

To ensure high quality for the questions, the solutions, the images, and the labels, we did two quality checks. Firstly, we generated all possible Depth 1 problems and manually verified their quality and correctness. Secondly, we asked 10 well-educated people to verify a total of 100 problems (from various depths and with various properties) and identify as many issues as possible with the questions, solutions, labels, or images. A list of the issues identified in this round are provided in Appendix E. All the raised issues were then fixed, and the process was repeated with 100 new examples to ensure no issues remained. Additionally, to get human performance on these problems, a separate set of four people solved 60 sampled problems (20 from each depth) and raised no issues, indicating another level of quality check for the generated dataset.

## 4. Experiments

We experiment with two state-of-the-art VLMs: PaLI (Chen et al., 2022b) and GPT4V (OpenAI, 2023), and a state-of-the-art LLM: PaLM 2 Large (Anil et al., 2023), in four settings: 1- zero-shot, 2- few-shot with chain-of-thought (CoT) prompting (Wei et al., 2022) (hereafter referred to as FS-CoT), where the CoT corresponds to the solution, 3- finetuning to directly predict the label (hereafter referred to as FT), and 4- finetuning to predict the solution/CoT (hereafter referred to as FT-CoT). We do the first experiment with GPT4V[3], the second with PaLM 2 Large and PaLI 55B (the largest PaLI model), and the last two experiments with PaLI 5B to keep the required computations manageable.

Following Methani et al. (2020) and Masry et al. (2022), we measure performance in terms of relaxed accuracy, where a prediction is considered correct if it is within $\delta$ percent of the golden label. We do this to accommodate for the slight variation in computations introduced due to the rounding strategy (e.g., due to the order of the computations). We empirically found $\delta = 3$ to be appropriate so we consider a prediction $p$ correct if $0.97 * label \leq p \leq 1.03 * label$. We remove from our dataset any example where the difference

---

[3]Based on the GPT4 responses, we notice that it uses zero-shot CoT (Kojima et al., 2022) under the hood.

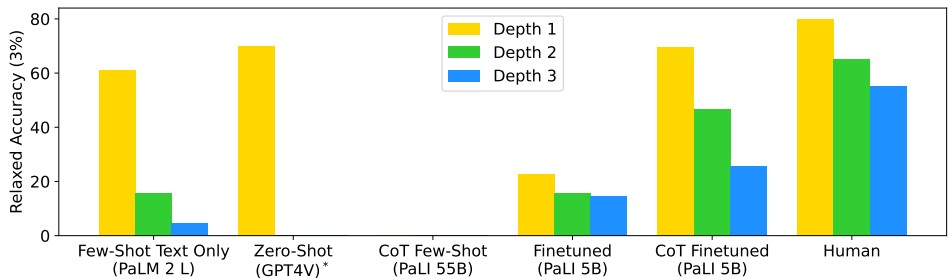

Figure 3: Model performances as a function of the depth of reasoning. **Note:** near-zero accuracies are not visible in the plot. *GPT4V results were obtained on a subset of randomly selected 10 examples per depth, and the correctness was determined manually.

between the label computed with and without rounding intermediate steps is more than 3%.

We provide results on subsets of our dataset with different properties. In each case, we generate 1000 examples randomly given the described parameters and report the results on those examples. We also generate a separate pool of train, validation, and fewshot examples for our experiments. The implementation details are presented in Appendix B.

### 4.1. Performance as a Function of Depth

Figure 3 represents the model results on examples with varying depths. Without finetuning, GPT4V can only solve Depth 1 examples, and the accuracy of the FS-CoT PaLI model is almost zero on all depths. In contrast, the text-only model can solve a portion of the Depth 2 and 3 problems as well. While the presence of the image should make the problem easier to understand and solve, this results hints that LLMs may be stronger in mathematical and multi-hop reasoning compared to their counterpart VLMs. Moreover, while finetuning helps VLMs learn to do some reasoning, as the depth of reasoning increases the performance drops monotonically and quite significantly.

Notice that FT-CoT outperforms FT substantially for all depths. While such improvements have been previously observed for reasoning with textual inputs (Suzgun et al., 2022), this result shows the importance of showing CoT to VLMs as well. This result also hints at the quality of the automatic solutions in GeomVerse.

We also measured human performance on our dataset by having 4 well-educated (but not necessarily expert in geometry) people solve a total of 20 problems per depth. The results show a stark gap between the best model performances and the human performance; we also observe that our problems can be challenging to solve even for humans. The mistakes made by humans where due to various issues including wrong/forgotten degree to radians conversion, using wrong formulas, and making wrong assumptions.

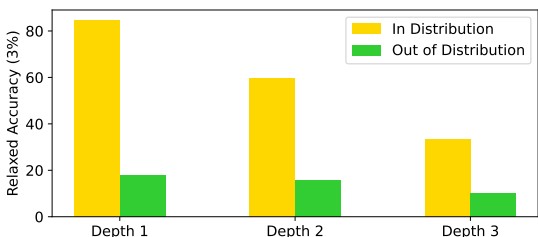

Figure 4: Measuring the generalization ability with in-distribution and out-of-distribution problems.

Table 2: Top-5 failure modes in the order of frequency.

| Few-Shot Text-Only Model | FT-CoT VLM Model (OOD) |
| --- | --- |
| Wrong proof planning | Wrong calculations |
| Wrong formula | Misunderstanding shapes |
| Wrong calculations | Wrong formula |
| Wrong assignment of values | Wrong proof planning |
| Hallucinating values | Wrong value assignment |

**Generalization:** We next measure how much the FT-CoT model (the best performing one across depths) can generalize to variations in the shapes. To this end, we finetune a model only on the following shapes: square, right triangle, trapezoid, semi-circle, rectangle plus equilateral triangle, rectangle minus semi-circle, and square minus circle. We then report the results separately for the test examples containing only these shapes (in-distribution) vs examples containing at least one new shape (out-of-distribution). Notice that all the left-out shapes have a similar (but not exact) counterpart shape in the training. The results are reported in Figure 4. As it can be observed, the performance goes significantly down for the out-of-distribution case.

Our depth and generalization results combined show that VLMs struggle with solving multi-hop geometry questions and reveals a crucial gap in their reasoning capabilities.

**Failure Analysis:** To understand the main failure modes of

the models, we manually verified 5 examples per depth for the FS-CoT text-only model and the FT-CoT model when tested on a combination of seen and unseen shapes. The main failure modes are presented in Table 2. Besides computation errors which have been previously observed as well for mathematical reasoning problems (Lewkowycz et al., 2022), we observe several other failure modes: 1- wrong proof planning (either wrong step order or disconnected steps), 2- wrong formulas (showing a gap in model knowledge), 3- misundestanding shapes in the case of VLMs (e.g., confusing sector with triangle), 4- wrong value assignment (e.g., assigning the value of a side to another side), and 5- hallucination (mostly hallucinating non-existent value). While proof planning is the most frequent failure mode of the text-only model, we notice that the FT-CoT model makes fewer planning errors.

**Correct Label = Correct Reasoning?** We next verify if the model produces a correct reasoning chain in the cases where it produces a correct final answer. Since re-using the reasoning chains produced by a model to further finetune it is becoming more prevalent (Zelikman et al., 2022; Huang et al., 2022; Magister et al., 2022), producing correct reasoning in the case of correct label is an important property of a model. To measure the reasoning accuracy, for the FS-CoT text-only and the FT-CoT models, we randomly selected up to 20 examples (upper-bounded by the number of correctly solved problems) from each of the depths where the model produced the exact label and verified manually if the produced reasoning chain is also correct. We also verified the examples for which the zero-shot model predicted the label correctly. For *Depth 1* examples, we observe that the reasoning chain is correct for the three models in all cases; for *Depth 2*, 20/20 have correct reasoning chains for the FS-CoT model and 19/20 have correct reasoning chains for the FT-CoT model, and for *Depth 3* examples, 8/9 examples have correct reasoning chains for the FS-CoT model and 16/20 for the FT-CoT model, with the errors mostly being on missed intermediate computations that were then replaced with correct numbers in later steps. This shows that the reasoning is mostly correct when the label is predicted correctly.

Due to the low performance of the zero-shot and FS-CoT PaLI models, hereafter we only experiment with the FS-CoT Text-Only and finetuned PaLI models.

### 4.2. Performance as a Function of Width

We generate *Depth 2* examples (medium difficulty in terms of depth) with $p_{branch}$=0, 0.5, and 1.0, and report the performances in Figure 5. We observe that while increasing the width negatively affects the performance in several cases (especially for the FS-CoT model) the amount of decrease

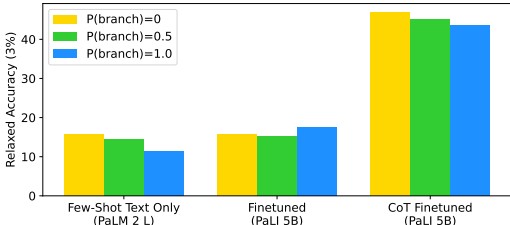

Figure 5: Model performances as a function of width. Models seem to be less affected by increasing the width of the reasoning.

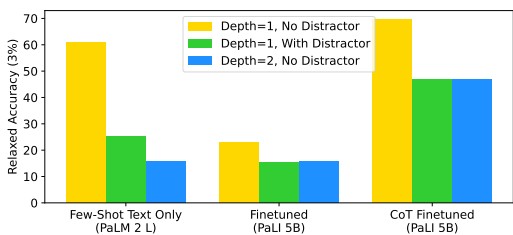

Figure 6: Model performance when distracting information is added to the question/image.

is substantially lower compared to the depth experiments[4]. The results hint at the ability of the models at learning to deal with higher width examples. This could be because the main added difficulty from higher width problems is that the model needs to solve more independent Depth 1 problems, on which they showed good performance according to Figure 3.

### 4.3. Distractors

We next measure how well models can deal with distracting information, a phenomenon which is common in real problems. We create a version of the Depth 2 problems where we provide the hidden value as input. This effectively turns the Depth 2 problem into a Depth 1 problem with some extra (distracting) shapes and values. The model performance is reported in Figure 6. Comparing Depth 1 results with and without distractors, the performance drops significantly for all models in presence of a distractor. Comparing Depth 1 with distractor and Depth 2 without distractor, while the text-only model has taken advantage of the value for the hidden element in some cases, for the finetuned VLM models the performance degrades to as low as that of the Depth 2 dataset.

---

[4]Part of the reason for this observation could be because we have only 30/68 formulas that have more than 1 connectable elements in their inputs and so even in the case where $p_{branch} = 1.0$, we still generate a number of examples that correspond to chains as opposed to trees.

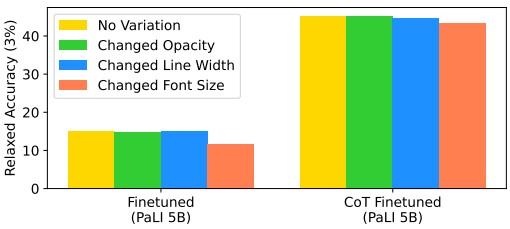

Figure 7: Measuring model sensitivity to low-level features of the images.

## 4.4. Sensitivity to low-level visual features

To measure how sensitive the VLM models are to the low-level visual features, we create separate test sets each varying in one low-level feature and measure the performance of the trained models on these new sets. Specifically, we experiment with changing the opacity of the shape colors, the line width of shape boundaries, and the font size of the texts on the images. In figure 7, we observe that the models are robust against opacity and line width, but not against font size changes.

## 4.5. Other Variations

Our experiments so far focus on general factors that may be present in many problems requiring reasoning on text and image. In Appendix A, we experiment with various other axes of difficulty that are more specific to geometry problems (including shapes, source of information, image annotation, adding variablized inputs, and decomposing performance based on question type).

## 5. Limitations & Risks

While GeomVerse covers a wide range of geometry questions, there are problems that cannot be produced using Algorithm 1 with our current set of shapes and formulas. The connection between Algorithm 1 and logical reasoning makes evident the class of problems that cannot be represented by the algorithm. In particular, let $\mathcal{P}$ be the class of geometry problems containing a tree of shapes where each shape is connected to its parent shape via a single side or a single (vertical) angle, and where the solution can be found by finding the values of the shared elements bottom-up on the tree. Algorithm 1 cannot generate any geometry problem that is not in $\mathcal{P}$. For example, let $ABC$ be a triangle, $D$ be a point on the $AC$ side dividing $ABC$ into two triangles $ABD$ and $ACD$, where some property of $ABC$ should be computed based on the properties of $ABD$ and $ACD$. This problem cannot be produced by Algorithm 1 as it does not correspond to a tree of connected shapes as described above. However, note that one can add such cases to our set of

non-standard shapes in a similar way we added the other non-standard shapes.

The problems in GeomVerse can be solved with a logical deduction procedure and may not require much creativity. For this reason, our evaluation should not be considered as measuring the creativity of the models in solving problems, but rather their ability in following a deduction procedure.

For our finetuning experiments, to make computations manageable, we used the small PaLI 5B model. Finetuning larger and more capable models such as Gemini (Team et al., 2023) or GPT4V (OpenAI, 2023) can provide more insight into the performance of the finetuned VLMs.

## 6. Conclusion

In this work, we procedurally generated a synthetic dataset of geometry reasoning questions that require multi-hop reasoning over both text and image. Through the lens of the geometry problems, we conducted a systematic analysis of various general and geometry-specific reasoning abilities of VLMs and found the gaps and strengths in their reasoning capabilities. Future work can verify the merit of finetuning models on synthetic geometry problems for improving their performance on real datasets. In an initial experiment, we measured the performance of the PaLI 5B model on Geometry3k with and without finetuning on GeomVerse and observed modest improvements (from almost 0 to almost 2 percent accuracy). We believe this is due to the difference in the visual and textual features of the Geomety3k and GeomVerse, as well as the poor generalization of PaLI to geometry problems beyond its training distribution. Better aligning the textual and visual features and using more powerful models can yield more gains.

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

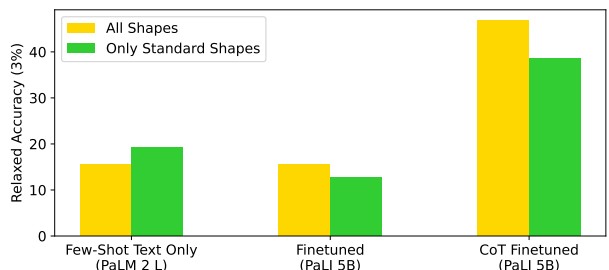

Figure 8: Comparing model performance when using only standard shapes vs when using all shapes. Overall, we do not see a big drop in the performance.

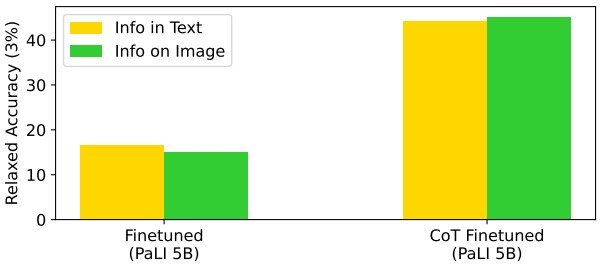

Figure 9: Model performances as a function of providing more information in the text or on the image.

## A. More Results: Other Axes of Difficulty

Besides the experiments in the main text, we also consider a number of other axes of difficulty for a systematic evaluation. In what follows, we describe these axes and present the experimental results. In Section D, we provide samples corresponding to each of the axes of difficulty.

### A.1. Standard vs Non-Standard Shapes

In Figure 2, we outlined the standard and non-standard shapes used in GeomVerse. Conceptually, it should be more difficult to solve problems involving non-standard shapes as they require more computations. We compare the performance of various models on problems that contain all shapes vs those that involve only standard shapes. To fix other axis of difficulty, we only consider depth 2 examples for this experiment where the problems are at a medium level of difficulty. The finetuned models are finetuned on all images in both cases. The results are in Figure 8. According to the results, we observe that while the FS-CoT model performs better on the standard shapes, this is not the case after finetuning. This shows that finetuning can teach the models to effectively deal with non-standard (but in-distribution) shapes.

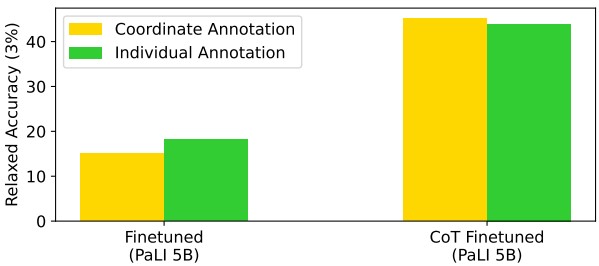

Figure 10: Model performances as a function of image annotation.

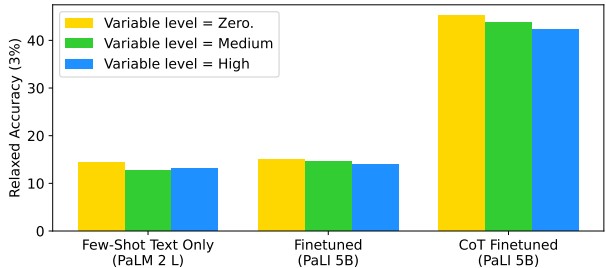

Figure 11: Model performances as a function of including variablized inputs in the question. The performance degrades as we include variables in the questions.

## A.2. More Info in Text or on Image

Some of the information can be provided either in the text of the question or on the image. For example, the degree of an angle can be provided in the image, or can be provided in the text. We generate examples where the information is given mostly in text and examples where it is given mostly on the image, and report model performances in Figure 9. For the FT model, we see that the former case results in lower accuracy which could be because in this case the model needs to first map those information to the elements in the image and then reason with them. FT-CoT almost closes the gap; this could be because the provided CoTs teach the model how to map information from text to image.

## A.3. Image Annotation

We consider two types of image annotation: 1- *individual annotation*: we refer to each side with a single lower-case letter, each angle with a Greek letter, and each shape with its (distinct) color, and 2- *coordinate annotation*: we assign upper-case letters to the coordinates on the image and refer to sides with the letters on the two coordinates, to angles with the three coordinates, and to shapes with all their coordinates. We generate a test set with *coordinate annotation* and another with *individual annotation* and report model performances on these two sets in Figure 10. The two models show different behaviour with the FT model performing slightly better on the individual annotation case, but the FT-CoT model slightly performing better on the coordinate annotation case.

## A.4. Variablized Inputs

Instead of providing the exact values of the input elements (e.g., the $\alpha$ angle is 30 degrees), it is common in geometry questions to provide a variablized version of them (e.g., the $\alpha$ angle is $2x + 1$) in which case one needs to first infer the value of the variable based on the given information and then use that to infer the value of an element. As an example, we can either directly provide two of the angles of a triangle as

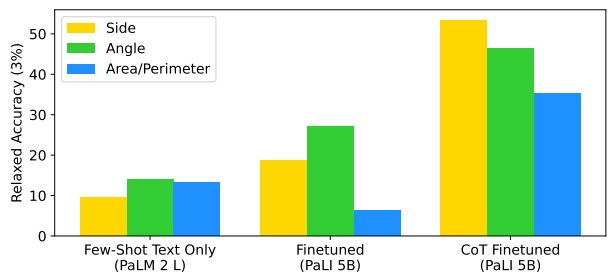

Figure 12: Model performances broken down by the question type.

input and ask for the third one, or we can provide variablized values for the three angles and ask for one of them. To generate variablized questions, when we use a formula $f$, instead of directly providing the values for $f^{(in)}$ as input and expecting the model to apply the formula to derive the value of $f^{(out)}$, we provide variablized values for (some of) the elements in $f^{(in)}$ and $f^{(out)}$ and expect the model to use the formula for deriving the value of the variable $x$ and use that to derive the numerical value of $f^{(out)}$. We selected $17/68$ of our formulas for which a variablized version of the problem only requires solving an extra 1-d linear equation. We then conducted an experiment where, whenever one of the 17 rules was selected during generation, we provide a variablized version of it with probability $\rho$. Figure 11 demonstrates the results for $\rho = 0$, $\rho = 0.5$ (corresponding to level = medium) and $\rho = 1.0$ (corresponding to level = high). We observe that as we include variablized inputs, the performance of the models degrade, especially for the FT-CoT model. This shows VLMs (and also LLMs) struggle to work with variables when solving geometry problems.

## A.5. Decomposing by Question Type

Our questions involve asking about the length of a side, the degree of an angle, or the area/perimeter of a shape. In Figure 12, we report model performances for each of these question types. We observe that the FS-CoT Text-Only

model performs almost equally across all three types, with slight preference for angle and area/perimeter questions. For the FT model, questions about angles are substantially easier, followed by questions about side. We conjecture that part of the reason for the high performance of the FT model on angle questions might be because the degree of an angle can be estimated from the figure without actually solving the problem. This could be in part validated by the results of the FT-CoT model, where the jump in accuracy is substantially higher for side and area/perimeter questions. In the case of FT-CoT, we see that side questions are easier than the other two; this may in part be because these questions involve easier arithmetic operations (e.g., some of the angle questions require computing *arcsin* which might be difficult for a pre-trained model).

## B. Implementation Details

For our finetuning experiments, we first generated a training set containing $10k$ examples and a validation set containing $2k$ examples. For each of the examples in these two sets, the parameters corresponding to different axes of difficulty discussed in the paper were set randomly to allow for a diverse set of examples in the train and validation sets. We then removed the (few) examples whose solution was identical to one of the solutions in one of the examples in our test sets. The same train and validation sets were used for all of our test sets.

We finetuned our model for $10k$ steps with a learning rate of $0.0005$ and a batch size of $128$, measured the model performance on the validation set every $2000$ steps, and reported the results on the test sets for the checkpoint achieving the best performance on the validation set.

For our fewshot experiments, we manually selected $4$ examples from the training set and used those examples as fewshot demonstrations across all experiments. These examples were selected to ensure many aspects of the test set are covered (e.g., to ensure there are examples at various depths, widths, with and without variables, with different question types, etc.).

**Rounding Errors:** Note that depending on how we round intermediate computations, the final answer can be slightly different. For example, consider the expression $\frac{2.26*3.14}{4}$. If we first multiply the numerator, round it and then divide by $4$ and round again, we will get $\frac{2.26*3.14}{4} = \frac{7.1}{4} = 1.78$. However, if we first do both computations and then round at the end, we will get $\frac{2.26*3.14}{4} = 1.77$. For this reason, we reported relaxed accuracy in our experiments to account for the differences in the way we computed the final results and the way the model may compute it.

## C. Sample Process for Algorithm 1

In Figure 13, we provide a visual demonstration of the process in Algorithm 1 for generating an example with Depth 3. In Step 1, we select a shape from our set of shapes and then select one of the formulas. The shape selected in this example is a rectangle and let the selected formula be to compute the area of a rectangle given its height and width; so $f_1^{(in)} = \{L_{AC}, L_{CD}\}$ and $f_1^{(out)} = \{A_{ABCD}\}$ where $L_{AC}$ and $L_{CD}$ represent the length of AC and CD and $A_{ABCD}$ represents the area of ABCD (note that we could also select $L_{BC}$ and $L_{AB}$ instead). We then select which element(s) from $f_1^{(in)}$ we will provide explicitly and which element(s) should be derived. Assume we decide to provide $L_{AC}$ explicitly and append other shapes to derive the value of $L_{CD}$. In this case, we assign a random value to $L_{AC}$ and provide it in the set of facts.

In Step 2, we need to select a shape where one of its sides is $CD$, and select a formula from which the length of this side can be derived. In the provided example, the selected shape is a right triangle and let us assume the selected formula is to compute a side of a right triangle given the hypotenuse and the opposite angle. So $f_2^{(in)} = \{L_{CE}, D_{CED}\}$ and $f_2^{(out)} = \{L_{CD}\}$, where $L_{CE}$ and $L_{CD}$ represent the lengths of the CE and CD sides and $D_{CED}$ represents the degree of the $CED$ angle. We then select which element(s) from $f_2^{(in)}$ we will provide explicitly and which element(s) should be derived. Assume both elements should be derived (corresponding to increasing the width of reasoning). So none of the elements will be added to the facts.

In Step 3, we need to select a shape where one of its sides is $CE$, and select a formula from which the length of this side can be derived. In the provided example, the selected shape is a semi-circle. Assume the formula is to compute the diameter of the semi-circle $L_{CE}$ given its perimeter $P_{SemiCircle}$. Since we want to generate Depth 3 examples, we add $P_{SemiCircle}$ to the set of facts.

In Step 4, we need to select a shape that can be connected to the $CED$ angle, such that $D_{CED}$ can be derived from that new shape. In the provided example, the selected shape is a supplementary angle, and the formula is that the sum of two supplementary angles is 180. We provide $D_{DEF}$ in the facts so $D_{CED}$ can be derived based on that.

Putting it all together, we get the rightmost shape in Figure 13. The facts include $\{L_{AC}, P_{SemiCircle}, D_{DEF}\}$ and the query is $A_{ABCD}$. Based on the rules we used, we can apply deduction to produce a solution as follows:

$$D_{DEF} \Rightarrow D_{DEC}$$
$$P_{SemiCircle} \Rightarrow L_{CE}$$
$$D_{DEC}, L_{CE} \Rightarrow L_{CD}$$

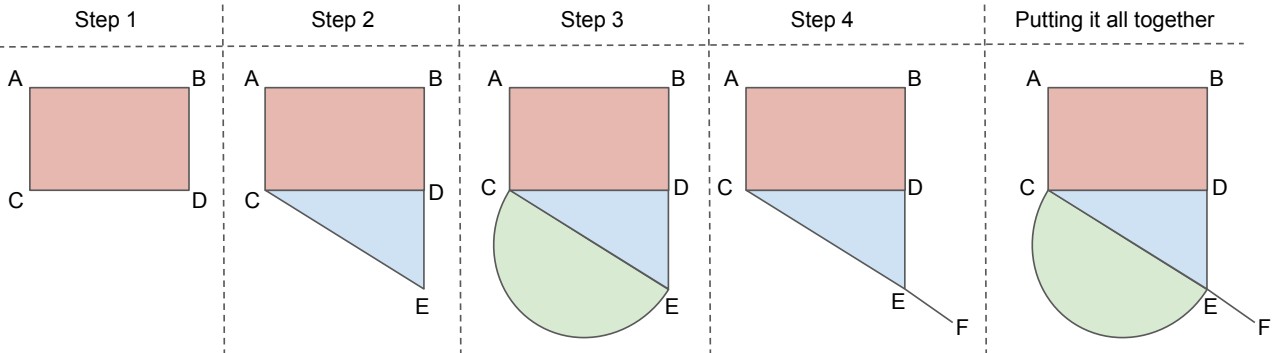

Figure 13: A visual demonstration of the process in Algorithm 1 for generating a example in Depth 3.

$$L_{CD}, L_{AC} \Rightarrow A_{ABCD}$$

To generate the question, we turn the facts (and the extra information needed to know such as the shapes, whether some angles are vertical or complementary, etc.) into a question using a template. We also provide the shape names when necessary. For Figure 13, for example, the elements whose values have to be provided as input are recorded during the generation process; this includes the length of AC, the measure of the DEF angle, and the perimeter of the semi-circle. We also take note of the other information that must be provided; this includes the fact that CDE is a right triangle and that DEF and DEC are complementary. We then use templates to turn each of these pieces of information into a textual format and concatenate them; we also textify the question using templates and append at the end. The final question will look like: *If the length of the AC side of the ABDC rectangle is 10, CDE is a right triangle, the DEF angle is 120 degrees, the DEF and the DEC angles are complementary, and the perimeter of the semi-circle is* 20, *compute the area of the ABDC rectangle.*

## D. Samples from GeomVerse

In this work, we experimented with several variations of GeomVerse. Here, we provide samples from these different variations to better illustrate how each test set looks like. The questions and solutions are provided in Tables 3 and 4 and the corresponding images are provided in Figure 14.

## E. Issues Found During Quality Check

As mentioned in the main text, the dataset went through multiple rounds of quality check. In what follows, we provide some of the examples of the issues found during the quality check by non-authors.

- **Text repetition:** In two cases, the quality checkers found the text of the question to be repetitive. This happened in the cases where, e.g., the question was *"the length of the AB side of the ABC triangle is 10, the length of the BC side of the ABC triangle is 6, the length of the AC side of the ABC triangle is 8"*. We updated our templates to remove repetitions.

- **Unnecessary information in the question:** An issue raised by multiple quality checkers was that we provided the value of $\pi = 3.14$ even when it was not used in the solution. We made sure we only provide it when needed. **Misprinting a formula in the solution:** In one case, a formula was misprinted in the solution where a squaring operation was missing (this did not affect the final result though because it was a misprint). This was fixed.

- **Unsolvable Variablized Inputs:** The quality checkers identified that when we provided variablized inputs, sometimes the problem became unsolvable. This happened, e.g., in the case where we provided an input such as *"the length of the three angles of a triangle are $x + 30$, $-2x + 60$ and $x + 45$"* where after summing the three values, $x$ disappeared.

- **Missing Coordinates:** In one case, one of the characters corresponding to an image in the coordinate was missing. We identified the root cause and fixed this.

- **Spacing issues:** Since the solutions were generated automatically, there were a number of cases where a space was either missing between two words or there were multiple spaces.

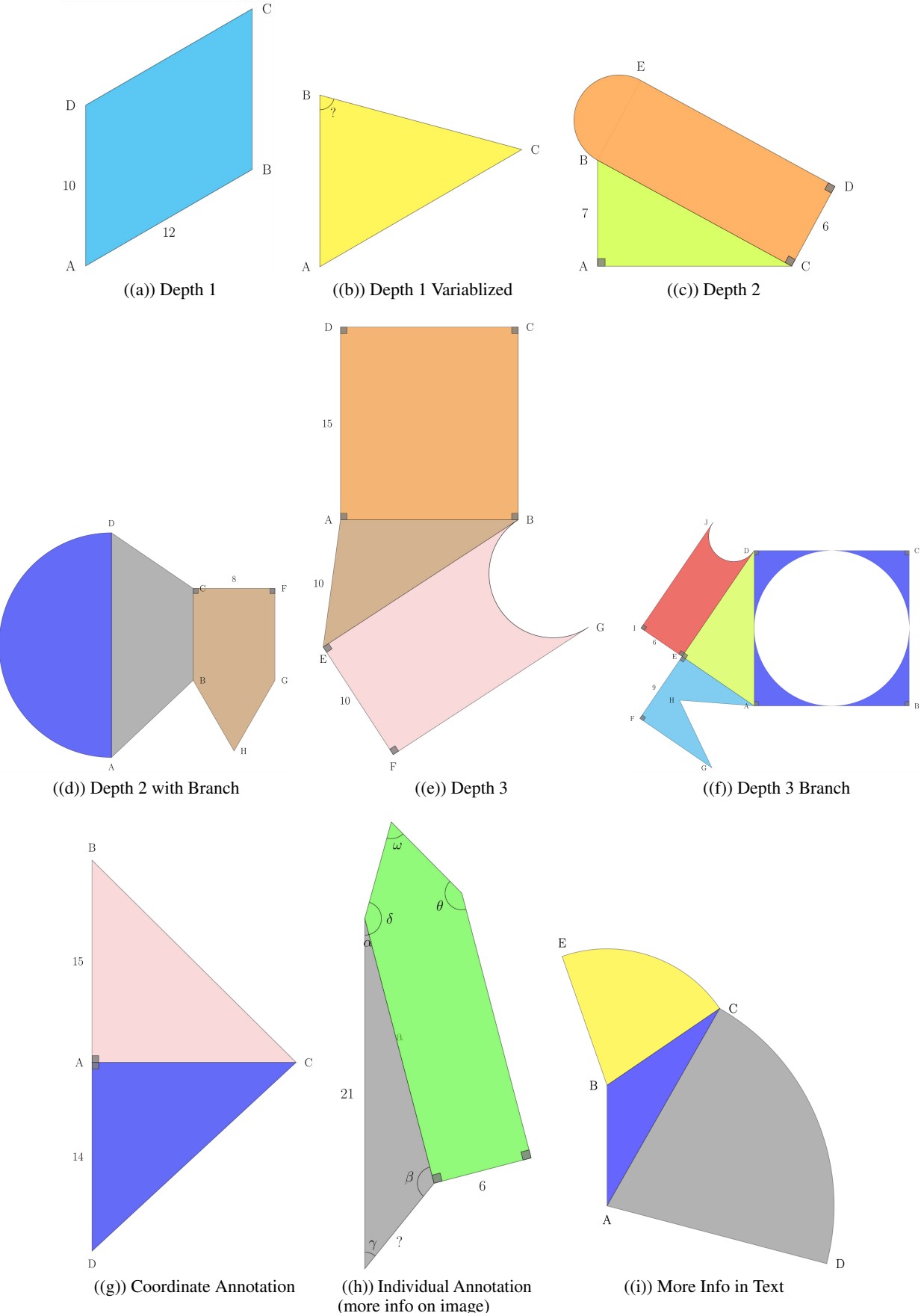

((a)) Depth 1

((b)) Depth 1 Variablized

((c)) Depth 2

((d)) Depth 2 with Branch

((e)) Depth 3

((f)) Depth 3 Branch

((g)) Coordinate Annotation

((h)) Individual Annotation (more info on image)

((i)) More Info in Text

Figure 14: Samples from our test sets.

Table 3: Questions and solutions corresponding to the images in Figure 14.

| Image index | Question | Solution |
|---|---|---|
| (a) | Compute the perimeter of the ABCD parallelogram. | The lengths of the AD and the AB sides of the ABCD parallelogram are 10 and 12, so the perimeter of the ABCD parallelogram is 2 * (10 + 12) = 2 * 22 = 44. Therefore the final answer is 44. |
| (b) | If the degree of the BAC angle is 2x + 50, the degree of the BCA angle is 2x + 35 and the degree of the CBA angle is 5x + 50, compute the degree of the CBA angle. | The three degrees of the ABC triangle are $2x+50$, $2x+35$ and $5x+50$. Therefore, $2x+50+2x+35+5x+50 = 180$, so $9x + 135 = 180$, so $9x = 45$, so $x = \frac{45}{9} = 5$. The degree of the CBA angle $= 5*5+50 = 75$. Therefore the final answer is 75. |
| (c) | If the BCDE shape is a combination of a rectangle and a semi-circle and the area of the BCDE shape is 102, compute the degree of the BCA angle. Assume $\pi = 3.14$. Round computations to 2 decimal places. | The area of the BCDE shape is 102 and the length of the CD side is 6, so (other side) $*6 + \frac{3.14*6^2}{8} = 102$, so (other side) * 6 = $102 - \frac{3.14*6^2}{8} = 102 - \frac{3.14*36}{8} = 102 - \frac{113.04}{8} = 102 - 14.13 = 87.87$. Therefore, the length of the BC side is $\frac{87.87}{6} = 14.65$. The length of the hypotenuse of the ABC triangle is 14.65 and the length of the side opposite to the BCA angle is 7, so the BCA angle equals $asin(\frac{7}{14.65}) = asin(0.48) = 28.69$. Therefore the final answer is 28.69. |
| (d) | If the length of the height of the ABCD trapezoid is 8, the area of the blue semi-circle is 189.97, the BCFGH shape is a combination of a rectangle and an equilateral triangle and the perimeter of the BCFGH shape is 42, compute the area of the ABCD trapezoid. Assume $\pi = 3.14$. Round computations to 2 decimal places. | The area of the blue semi-circle is 189.97 so the length of the AD diameter can be computed as $\sqrt{8 * \frac{189.97}{\pi}}) = \sqrt{\frac{1519.76}{\pi}} = \sqrt{484.0} = 22$. The side of the equilateral triangle in the BCFGH shape is equal to the side of the rectangle with length 8 so the shape has two sides with equal but unknown lengths, one with length 8, and two triangle sides with length 8. The perimeter of the BCFGH shape is 42 so 2 * (unknown side) + 3 * 8 = 42. So 2 * (unknown side) = 42 - 24 = 18, and the length of the BC side is $\frac{18}{2} = 9$. The lengths of the AD and the BC bases of the ABCD trapezoid are 22 and 9 and the height of the trapezoid is 8, so the area of the trapezoid is $\frac{1}{2} * (22 + 9) * 8 = \frac{1}{2} * 31 * 8 = 124$. Therefore the final answer is 124. |
| (e) | If the perimeter of the ABE triangle is 42, the BEFG shape is a rectangle where a semi-circle has been removed from one side of it and the perimeter of the BEFG shape is 62, compute the perimeter of the ABCD rectangle. Assume $\pi = 3.14$. Round computations to 2 decimal places. | The diameter of the semi-circle in the BEFG shape is equal to the side of the rectangle with length 10 so the shape has two sides with equal but unknown lengths, one side with length 10, and one semi-circle arc with diameter 10. So the perimeter is $2 * UnknownSide + 10 + \frac{10*\pi}{2}$. So $2 * UnknownSide + 10 + \frac{10*3.14}{2} = 62$. So $2 * UnknownSide = 62 - 10 - \frac{10*3.14}{2} = 62 - 10 - \frac{31.4}{2} = 62 - 10 - 15.7 = 36.3$. Therefore, the length of the BE side is $\frac{36.3}{2} = 18.15$. The lengths of the AE and BE sides of the ABE triangle are 10 and 18.15 and the perimeter is 42, so the lengths of the AB side equals $42 - 10 - 18.15 = 13.85$. The lengths of the AD and the AB sides of the ABCD rectangle are 15 and 13.85, so the perimeter of the ABCD rectangle is $2 * (15 + 13.85) = 2 * 28.85 = 57.7$. Therefore the final answer is 57.7. |

Table 4: Continuing Table 3.

| Image index | Question | Solution |
|---|---|---|
| (f) | If the ABCD shape is a square where a circle has been removed from it, the AEFGH shape is a rectangle where an equilateral triangle has been removed from one side of it, the perimeter of the AEFGH shape is 48, the DEIJ shape is a rectangle where a semi-circle has been removed from one side of it and the area of the DEIJ shape is 78, compute the area of the ABCD shape. Assume $\pi = 3.14$. Round computations to 2 decimal places. | The side of the equilateral triangle in the AEFGH shape is equal to the side of the rectangle with length 9 and the shape has two sides with equal but unknown lengths, one with length 9, and two triangle sides with length 9. The perimeter of the shape is 48 so $2*OtherSside+3*9 = 48$. So $2*OtherSide = 48 - 27 = 21$ and the length of the AE side is $\frac{21}{2} = 10.5$. The area of the DEIJ shape is 78 and the length of the EI side is 6, so $OtherSide * 6 - \frac{3.14*6^2}{8} = 78$, so $OtherSide * 6 = 78 + \frac{3.14*6^2}{8} = 78 + \frac{3.14*36}{8} = 78 + \frac{113.04}{8} = 78 + 14.13 = 92.13$. Therefore, the length of the DE side is $92.13/6 = 15.35$. The lengths of the AE and DE sides of the ADE triangle are 10.5 and 15.35, so the length of the hypotenuse (the AD side) is $\sqrt{10.5^2 + 15.35^2} = \sqrt{110.25 + 235.62} = \sqrt{345.87} = 18.6$. The length of the AD side of the ABCD shape is 18.6, so its area is $18.6^2 - (\frac{\pi}{4})*(18.6^2) = 345.96 - 0.79 * 345.96 = 345.96 - 273.31 = 72.65$. Therefore the final answer is 72.65. |
| (g) | If the area of the ACD right triangle is 106, compute the area of the ABC right triangle. Round computations to 2 decimal places. | The length of the AD side in the ACD triangle is 14 and the area is 106 so the length of the AC side = $\frac{106*2}{14} = \frac{212}{14} = 15.14$. The lengths of the AC and AB sides of the ABC triangle are 15.14 and 15, so the area of the triangle is $(15.14 * 15)/2 = 227.1/2 = 113.55$. Therefore the final answer is 113.55. |
| (h) | If the perimeter of the gray triangle is 44, the green shape is a combination of a rectangle and an equilateral triangle and the area of the green shape is 114, compute the length of the side of the gray triangle marked with question mark. Round computations to 2 decimal places. | The area of the green shape is 114 and the length of one side of its rectangle is 6, so (other side) $*6 + \frac{\sqrt{3}}{4} * 6^2 = 114$, so (other side) $*6 = 114 - \frac{\sqrt{3}}{4} * 6^2 = 114 - \frac{1.73}{4} * 36 = 114 - 0.43*36 = 114 - 15.48 = 98.52$. Therefore, the length of the side marked with letter "a" is $98.52/6 = 16.42$. The lengths of two sides of the gray triangle are 21 and 16.42 and the perimeter is 44, so the lengths of the side marked with "?" equals $44 - 21 - 16.42 = 6.58$. Therefore the final answer is 6.58. |
| (i) | If the perimeter of the ABC triangle is 33, the degree of the CAD angle is 75, the area of the DAC sector is 157, the degree of the EBC angle is 75 and the area of the EBC sector is 56.52, compute the length of the AB side of the ABC triangle. Assume $\pi = 3.14$. Round computations to 2 decimal places. | The CAD angle of the DAC sector is 75 and the area is 157 so the AC radius can be computed as $= \sqrt{157/((75/360) * \pi)} = \sqrt{157/(0.21 * \pi)} = \sqrt{157/0.66} = \sqrt{(237.88)} = 15.42$. The EBC angle of the EBC sector is 75 and the area is 56.52 so the BC radius can be computed as $= \sqrt{56.52/((75/360) * \pi)} = \sqrt{56.52/(0.21 * \pi)} = \sqrt{56.52/0.66} = \sqrt{85.64} = 9.25$. The lengths of the AC and BC sides of the ABC triangle are 15.42 and 9.25 and the perimeter is 33, so the lengths of the AB side equals $33 - 15.42 - 9.25 = 8.33$. Therefore the final answer is 8.33. |

