# OpenReview forum: "GeomVerse: A Systematic Evaluation of Large Models for Geometric Reasoning"
_ICML.cc/2024/Workshop/AI4MATH — ICML 2024 Workshop AI4MATH Poster_

### Official Review · Reviewer_BePn · 2024-06-12

**Rating:** 8
**Confidence:** 4

**Summary:**

This paper has a very clear structure with details. It also has relatively good literature review. Moreover, the proposed method to generate school geometric problems is systematic and innovate. In general it is a good paper and is worthy a contributed talk or good poster at least.

**Questions:**

1. In Section 3.2, it is mentioned that the majority of the geometric reasonings in this study are based on some values and the logics between them. The sample questions all have a clear numerical answer. Have the authors considered creating and testing geometric questions whose answers are not a determined number? The question descriptions may not include specific values.

2. In Section 4.1, when discussing "Generalization", it is discussed that a subgroup of shapes are used to finetune the model; does the choice of this subgroup have impact on the experiment?

**Reasons To Accept:**

The idea and algorithm proposed in this paper for generating school geometric level problems are relatively new; the idea is clear and effective. Also, the authors did a very careful analysis of their proposed algorithm and the resulting dataset, including the issues they met. This is also very good experience to share and study.

**Reasons To Reject:**

I would recommend this paper in general, at least for a contributed talk or a very good poster. The only little concern here is that the topic discussed in this paper  seems not one of the central themes of the conference, though related.

---

### Official Review · Reviewer_cmL9 · 2024-06-12

**Rating:** 8
**Confidence:** 4

**Summary:**

This paper presents a detailed assessment of VLMs using geometry problems. It introduces the GeomVerse dataset, a procedurally generated set of geometry questions designed to test multi-hop mathematical reasoning involving both text and images. The authors demonstrate that while VLMs can handle simple reasoning tasks, they struggle with more complex problems requiring deeper reasoning and generalization. Key findings include the models' limitations in dealing with higher-depth problems and distractors, highlighting the need for further advancements in VLMs for them to be effective in educational applications and beyond.

**Questions:**

N/A

**Reasons To Accept:**

1. **Novel Dataset**: The paper introduces GeomVerse, a procedurally generated dataset that is both unique and innovative. It systematically evaluates VLMs on geometric reasoning tasks, combining text and images, which addresses a significant gap in existing research datasets. This novel dataset allows for controlled difficulty levels across multiple axes, enabling a thorough and nuanced assessment of model capabilities.

2. **Comprehensive Experiments and Analysis**: The paper provides an exhaustive evaluation of state-of-the-art VLMs through various experiments. The authors meticulously analyze model performance across different depths and widths of reasoning, assess generalization abilities, and examine the impact of distractors. Additionally, they explore the sensitivity of models to low-level visual features and different types of image annotations. This thorough experimental setup and detailed analysis offer valuable insights into the current limitations and strengths of VLMs in geometric reasoning, making the study a significant contribution to the field.

**Reasons To Reject:**

N/A

---

### Meta-Review · Area_Chair_QLER · 2024-06-13

**Recommendation:** Accept (Oral, top 2)
**Confidence:** 4

**Metareview:**

Nice paper. It proposes a new dataset of geometry problems to benchmark the performance of visual language models (VLMs) with clear presentation and comprehensive empirical study. It could serve to the AI for Math community in the future as a common benchmark in this direction.

---

### Decision · Program_Chairs · 2024-06-13

Accept (Poster)